# Seroprevalence Assessment and Risk Factor Analysis of *Toxoplasma gondii* Infection in Goats from Northeastern Algeria

**DOI:** 10.3390/ani14060883

**Published:** 2024-03-13

**Authors:** Abdeldjalil Dahmane, Daniela Almeida, Nassiba Reghaissia, Djamel Baroudi, Houssem Samari, Amine Abdelli, AbdElkarim Laatamna, João R. Mesquita

**Affiliations:** 1Higher National Veterinary School, Issad Abbas Street, Bab Ezzouar, Algiers 16000, Algeria; a.dahmane@etud.ensv.dz (A.D.); dbaroudi7@hotmail.com (D.B.); 2Laboratory of Exploration and Valorization of Steppic Ecosystems, Faculty of Nature and Life Sciences, University of Djelfa, Moudjbara Road, BP 3117, Djelfa 17000, Algeria; 3Department of Agronomic Sciences, Faculty of Exact Sciences, Nature and Life Sciences, University of Biskra, Biskra 07000, Algeria; 4School of Medicine and Biomedical Sciences (ICBAS), University of Porto, 4050 Porto, Portugal; dgomesalmeida23@gmail.com; 5Institute of Veterinary and Agronomic Sciences, University of Souk Ahras, Annaba Road, Souk Ahras 41000, Algeria; n.reghaissia@univ-soukahras.dz; 6Faculty of Sciences, University of M’sila, M’sila 28000, Algeria; houssem.samari@univ-msila.dz; 7Department of Agricultural Sciences, University of Bouira, Bouira 10000, Algeria; abdelliamine@hotmail.fr; 8Epidemiology Research Unit (EPIUnit), Instituto de Saúde Pública da Universidade do Porto, 4050 Porto, Portugal; 9Laboratory for Integrative and Translational Research in Population Health (ITR), 4050 Porto, Portugal

**Keywords:** *Toxoplasma gondii*, caprine, Algeria, antibody detection, epidemiology

## Abstract

**Simple Summary:**

Toxoplasmosis, a zoonotic protozoan disease caused by the pathogen *Toxoplasma gondii*, can infect almost all homoeothermic animals including humans. *Toxoplasma gondii* is responsible for reproductive disorders in ruminants such as goats which could have an important role as a source of parasite transmission to humans. Data on seroepidemiology of *T. gondii* in Algeria is scarce, and almost null in the rural areas. This study was carried out to determine seroprevalence and to identify the associated risk factors for *T. gondii* infection in goats from four provinces in northeastern Algeria. A total of 460 serum samples were tested from 72 herds. A total of 245 samples showed the presence of anti-*T. gondii* IgG antibodies, representing 53.26% of the goats and 94.44% of herds, indicating they were positive for *T. gondii*. Factors such as the presence of a water source in the pasture, number of cats, and number of abortions were significantly linked to the high seropositivity of *T. gondii*. This investigation suggests that *T. gondii* is widespread in the goat population in Algeria and the transmission of the parasite to humans could potentially increase via meat consumption.

**Abstract:**

*T. gondii* is the causal agent of toxoplasmosis, a worldwide zoonotic disease relevant in human and veterinary medicine. In Algeria, few reports focused on the presence and circulation of this parasite in the local goat population. The aim of the survey was to evaluate toxoplasmosis seroprevalence and associated risk factors. Sera from 460 goats reared on 72 farms in northeastern Algeria were collected and tested for IgG antibodies to *T. gondii* by an indirect ELISA. To identify risk factors, a linear regression analysis of the variables was performed. Anti-*T. gondii* antibodies were found in 94.44% (68/72; 95% CI: 73.34–119.73) of goat farms and in 53.26% (245/460; 95% CI: 46.80–60.36) at the individual level. The multivariable analysis showed that seasonal pasture (OR = 3.804; 95% CI: 3.321–4.358; *p* = 0.003), presence of water source in pasture area (OR = 4.844; 95% CI: 1.942–7.789; *p* = 0.0004), use of anthelminthics (OR = 2.640; 95% CI: 1.592–3.146; *p* = 0.036), number of cats, hygiene, proportion of abortions, number of abortions in the last year, year of sampling, region, and season were the variables significantly associated with *T. gondii* seropositivity. Abortions in goat herds seem to be related to *T. gondii* exposure, thus it is crucial to undertake measures and strategies to reduce, control, and prevent toxoplasmosis infection in goats, and thereby in humans, from Algeria.

## 1. Introduction

*Toxoplasma gondii* is an obligate intracellular Apicomplexa parasite that can infect almost all warm-blooded animals, including farm animals, birds, sea mammals, and humans, with a variety of clinical manifestations [1,2]. The life cycle of this important zoonotic protozoan implies domestic cats, as well as other felids, as definitive hosts, and a wide range of mammals, including mainly herbivorous ones, and humans as intermediate hosts. Domestic cats and other felidae excrete via feces non-sporulated oocysts which are environmentally resilient [1]. The oocysts become infectious after sporulation, occurring when contacting the environment for 1 to 5 days [3]. Tachyzoites, the rapid-growing life stage of the parasite, and bradyzoites, which are the life stage present in tissue cysts, are two further infectious forms of *T. gondii* other than sporulated oocysts [4,5]. Generally, intermediate hosts acquire infection through ingestion of food or water contaminated by sporulated oocysts. Furthermore, ingestion of uncooked or undercooked meat as well as other animal products containing infectious parasitic stages (tissue bradyzoites) represents an important mode of transmission, specifically in humans [3].

Toxoplasmosis is considered one of the major constraints on livestock production worldwide, causing serious reproductive losses such as abortion, stillbirth, and a fall in milk production [6]. Furthermore, toxoplasmosis is of considerable public health importance, making it one of the most common zoonotic parasitic infections in humans worldwide [7]. Mostly, the infection seems to be asymptomatic in immunocompetent individuals, while causing serious problems in others, specifically in neonates and immunocompromised persons [1].

Of particular interest, toxoplasmosis represents a serious problem in the breeding of goats. It has been reported that 3.3% to 27.2% of goat abortions around the world were associated with *T. gondii* infection [8]. Fetal and newborn mortality rates in infected flocks can exceed 50%, with only minor losses in non-clinical cases [9]. Goats are most likely affected when they consume feed or water that has been contaminated with cat feces [8]. In this regard, *T. gondii* infection is a significant water-borne disease, and it may be influenced by ecosystem fragmentation, poor water quality, and changes in water flow [7,10].

Goats are one of the intermediate host species where *T. gondii* is vertically transmitted, with tachyzoites contributing to transplacental transmission typically during the acute phase of infection or after a reactivated chronic infection [1]. *Toxoplasma gondii* infection in goats was mostly investigated by the assessment of seroprevalence as compared to the detection of the parasite or its DNA in infected tissues as well as in blood and milk. *T. gondii* DNA has been recovered in goats that were slaughtered across the world, and living bradyzoites have also been found in the muscles of naturally infected goats [8,11]. The consumption of raw goat milk containing tachyzoites as well as undercooked meat containing bradyzoites is thought to be a significant source of infection in humans, with particular impact on pregnant women [12,13]. The role of goat’s raw milk as a source of human infections remains uncertain due to the absence of solid evidence showing the presence of viable *T. gondii* [8].

In most developing nations, at least 30% of the population has *T. gondii* IgG antibodies [1]. High prevalence rates of human infection have been documented in recent years in North Africa, including Morocco (51%) [14] and Algeria (40.97% to 47.8% in pregnant women) [15,16].

*Toxoplasma gondii* is a widespread parasite that affects small ruminants all over the world. Several investigations have investigated the seroprevalence of *T. gondii* antibodies in goats and the potential associated risk factors in various countries, indicating infection rates ranging from 4.6 to 73.3% depending on the geographical area, the diagnostic approach, and the requested cut-off titer [8,17,18].

In different countries, breeding of goats is important for the economy, specifically for milk and meat production [19]. In Algeria, livestock husbandry, particularly small ruminant breeding, represents an important agricultural activity in the rural high plateaus and steppe areas. One of the most important areas for livestock production and agriculture is in northeastern Algeria, where goat farming is an additional economic activity that is primarily characterized by small, non-racial herds and tried-and-true specialties in an often semi-extensive system. The role of small ruminants, including goats, as a significant reservoir of zoonotic parasites such as *T. gondii* in rural areas of Algeria is poorly documented. Relatively limited epidemiological data are available in Algeria on toxoplasmosis in goats as well as in other animal species and humans [20,21]. The frequency of toxoplasmosis in goats in the northeastern region of Algeria has been reported in only one study so far [20]. To support current available data, the present study was conducted to provide the latest information on the seroprevalence of *T. gondii* infection in goats in northeastern Algeria and offer an up-to-date summary of potential associated risk factors using detailed statistical analysis.

## 2. Materials and Methods

### 2.1. Study Area and Environment

The present study was carried out in four provinces including Mila, Constantine, Guelma, and El Taref, which are located in northeastern Algeria (Figure 1).

Mila lies inland, about 82 km from the Mediterranean coast. The district is characterized by a varied relief and presents two large, distinct zones: to the north, the mountains, and to the south, the plains and highlands, with an area of 3481 km^2^. The region has a humid Mediterranean climate, with hot, dry summers and cold, wet winters. The annual rainfall is approximately 550 mm with a relative humidity of 70%, and the annual temperature varies between 8 °C and 24 °C [22]. Constantine is situated on a plateau at 698 m above sea level; the area has a humid climate with hot, dry summers and cold, moist winters and is characterized by an annual rainfall of 600 mm, an annual temperature range of 10 °C to 25 °C, and 60% annual relative humidity [22]. The territory of Guelma province is characterized by a sub-humid climate in the center and the north and a semi-arid climate towards the south. This climate is mild and rainy in the winter and warm in the summer. Thus, the annual temperature varies between 12.5 °C and 25.5 °C, accompanied by 750 mm of annual rainfall and a relative humidity of 58.3%. El Taref province is located in the far northeast of Algeria, close to the Tunisian border. The climate is generally humid, with an important annual rainfall of 1000 mm, a high annual relative humidity of 77%, and an annual temperature ranging from 4 °C to 23 °C [22]. Livestock husbandry in these four provinces consists of a mixture of breeding cattle and small ruminants. According to data from the last years (2019–2020) from the Algerian Ministry of Agriculture, approximatively 73,658, 34,623, 35,136, and 11,365 goat heads are raised in Guelma, El Taref, Mila, and Constantine, respectively [23].

### 2.2. Study Design and Target Population

A cross-sectional study was conducted. An appropriate number of goats were sampled by a simple random sampling method. In selecting the municipalities and properties that participated in the study, the division of the state, ease of access, convenience, and availability of producers were taken into account. The goats were randomly selected from males and females, apparently healthy with different production patterns, and aged over three (03) months. The number of goats to be taken from each farm was defined based on the total number of animals. A representative sample of at least 10% of all individuals on each farm visited was achieved.

The required sample size was calculated according to the following formula [24] with an expected prevalence of 63% [20], an expected error of 5%, and assuming a 95% confidence interval:N = [Z2 × P (1 − P)]/d2
where:N is the number of samples to be collected in the study;Z is the value of the normal distribution for the confidence interval of 95% [Z= 1.96];P is the expected prevalence;d is the absolute error or required precision of ±5% for a 95% confidence interval (0.05).

A minimum of 358 samples was required. Seventy two herds were randomly selected and the herd sizes ranged from 5 to 50 heads.

At the individual level, the sample size was determined for each flock for eventual detection of *T. gondii* antibodies. The calculations were performed according to the formula commonly used in veterinary epidemiological surveys [24]:n = ([1 − (1 − p) 1/d] × [N − (d/2)]) + 1
where:n is the size of the sample in each flock;p is the probability of detection of at least one seropositive goat in a herd determined at 95%;N is the size of the flock;d is the number of seropositive goats in the herd (it was calculated assuming that within-herd prevalence equals 10%).

### 2.3. Data Collection and Sampling

During visits to the properties, a structured questionnaire was administered to each farmer under the supervision of the principal investigator to assess the risk factors associated with *T. gondii* infection. The questionnaires consisted of several closed questions about the gender, age, characteristics of the herd, breeding purpose, management system, reproductive disorders such as abortion and stillbirth, and cat-related factors [8,9,17].

Blood samples were collected from September 2020 to March 2023, totaling 460 goats from about 72 farms in fifteen municipalities in four provinces (Mila, Constantine, Guelma, and El Taref). Blood samples (2–3 mL) were collected from the venipuncture of the jugular with sterile 40 × 12 needles in a vacuum tube (Vacutainer^®^, VacuTube, Algiers, Algeria), without anticoagulant, and transported under refrigeration to the laboratory. They were centrifuged at 2000× *g* for 10 min to obtain serum, stored in Eppendorf tubes, and frozen at −20 °C until serological analysis.

### 2.4. Serological Analysis

All sera samples (*n* = 460) were tested for *T. gondii* IgG antibody detection using the ID Screen Toxoplasmosis Indirect ELISA Multi-species kit (ID Screen, ID.VET. Innovative Diagnostics, Montpellier, France), according to the manufacturer’s instructions. This is an indirect ELISA that uses the native P30 (SAG1) antigen and the anti-multi-species conjugate as the secondary antibody. This configuration is effective for identifying *T. gondii*-specific antibodies present in the serum of ruminants, pigs, dogs, and cats, as well as in milk and meat juice [25]. Moreover, the use of the SAG1 antigen in this kit might provide greater specificity than tests using a whole tachyzoite antigen [26]. Positive and negative controls (supplied in the kit) were tested in duplicate, as per instructions. Results were expressed as a percentage of the optical density (OD) reading of the test, calculated as %OD = 100 × (OD sample − OD Negative Control)/(OD Positive Control − OD Negative Control). The samples were considered positive if they had a value ≥50%, doubtful for values between 40% and 50%, and negative if ≤40%.

### 2.5. Statistical Analysis

Data obtained from the questionnaire survey and ELISA analysis were recorded and coded in a Microsoft Excel spreadsheet (version 16.0) (Microsoft Corporation, Washington, USA), and were used to calculate seroprevalence values specific to the population and geographic locations (provinces). Descriptive analysis was applied to describe the study population in relation to risk factors and variables. We evaluated 139 variables of plausible risk factors (57), including gender of the animals, age, contact with other animal species, water source, management practices, presence of cats, number of cats, flock size, history of abortions, and proportion of abortions in the flock. The association of the assumed risk factors for *T. gondii* seropositivity was analyzed by univariate analysis that was performed using Pearson’s chi-square test and, when necessary, Fisher’s exact test. Variables with a *p*-value < 0.05 in univariate analysis were offered to the multivariate model analysis via generalized linear mixed model fit by maximum likelihood, considering the herd as the random effect depending on the result of the ELISA test. The explanatory variables considered fixed effects in the model were those presenting statistical significance. The full model was reduced by automatic model selection based on a finite sample AIC. The process was performed using R software (version 3.5.1; R Foundation for Statistical Computing, Vienna, Austria) via RStudio (version 1.1.383, RStudio Inc., Boston, MA, USA). In the final multivariate linear regression analysis, variables with a *p* < 0.05 were considered significant. The confidence interval was established at 95%.

## 3. Results

A total of 460 serum samples from goats were collected in different agroecological areas of northeastern Algeria and tested by indirect ELISA to detect IgG antibodies against *T. gondii*. Out of the 460 samples, 245 samples were found to be positive, making the overall prevalence in domestic goats 53.26% (245/460; 95% CI: 46.80–60.36) as shown in Table 1. At the herd level, 68/72 (94.44%; 95% CI: 73.34–119.73) had at least one animal serologically positive for *T. gondii*.

The results regarding the general characteristics of goats exhibited the fact that the rate of seropositive goats with toxoplasmosis was not affected by factors such as gender and age (*p* > 0.05), and showed that animals with different body conditions were equally infected by *T. gondii* (*p* = 0.59). These results are available in the Appendix A. The seroprevalence of toxoplasmosis in pregnant goats (56.19%; 95% CI: 42.77–72.48) was found to be double that in non-pregnant goats (25%; 95% CI: 06.81–64.01), and that finding was statistically significant (*p* = 0.02). Pregnancy was observed to be a significant risk factor, and exhibited a significant increase in the risk of suffering from the disease as compared to a non-pregnant animal. A pregnant animal was 0.25 times more likely to be infected with the parasite than a non-pregnant animal. Physiology status of male goats seems to impact the risk of exposure to the parasite, which was more prevalent in breeder males (64.18%; 95% CI: 46.45–86.45) than non-breeder males (46.15%; 95% CI: 33.26–62.39) with a *p*-value equal to 0.02.

Univariate analysis showed that most of the factors related to the management system and herd characteristics were statistically non-significant (*p* > 0.05) and did not influence the risk to *T. gondii* exposure. Pasture (No/Yes) had a negative correlation with *T. gondii* prevalence, where goats raised permanently within farms were significantly more seropositive and more exposed by 0.37 times then the ones grazing in pasture (*p* = 0.02). In addition, pasture frequency highly impacted the exposure rate to the parasite with significantly association (*p* = 0.001). Regarding transhumance as a variable, the results revealed that it was a risk factor (*p* = 0.04) but with a limited effect on exposure to *T. gondii* related to a low value of OR (OR = 0.28) between categories of animals with and without transhumance practice. Few goats were submitted to transhumance, and only 25% (95% CI: 05.16–73.06) of them had antibodies against *T. gondii* and appeared to be less exposed. 

Water- and feed-related factors had no significant chance to be positively correlated with seroprevalence variations regarding toxoplasmosis. A factor that appeared to be potentially implicated in the increasing of animals’ exposure to *T. gondii* was the presence of a water source in the pasture area (*p* < 0.001). Goats grazing in areas associated with a water source showed a high risk of being seropositive (OR= 2.24 times and *p* < 0.001). The seroprevalence of goats was also influenced by the presence of dogs in herds, where goats were less exposed to the parasite (50.91%; 95% CI: 44.02–58.58) than those living in herds without dog cohabitation (64.94%; 95% CI: 48.20–85.61), and that was significantly different (*p* = 0.02). The presence of cats on farms was not considered as a risk factor until *p*-value was higher than 0.05, and under the category of cat-related factors, only the number of cats seemed to significantly impact the rate of seropositivity against *T. gondii* (*p* = 0.031). In herds where the presence of cats was recorded, the presence of three to four cats in the herd increased the risk of the disease to 4.43 times that of the situation where only one to two cats were present. Furthermore, herd areas where more than five cats were being raised had OR = 0.77 compared to cases where only three or four cats were being raised. 

The use of anthelminthic drugs notably decreased the percentage of *T. gondii* infection in goats (*p* = 0.01) by 0.61 times. Application of vaccines against other pathogens was associated with a high prevalence of toxoplasmosis (OR = 1.49), the difference being statistically significant (*p* = 0.03). Hygiene was strongly associated with the prevalence of anti-*T. gondii* IgG antibodies, as the rate was 48.08% (49/102; 95% CI: 35.54–63.51) in farms with bad hygiene conditions and 37.10% (23/62; 95% CI: 23.52–55.66) in those with good hygiene (*p* = 0.004). Goats living in good hygiene conditions had reduced risk of infection by the parasite with OR = 1.56. Similarly, goats in herds with a different range of abortion proportions had notably significantly different rate of exposure (*p* = 0.003). The rate of seropositivity highly increased with the proportion of abortions, presenting a positive correlation with a value of odds ratio ranging from 0.55 to 0.61. The link between *T. gondii* prevalence rate and the number of abortions in the last year was assessed and revealed to be significant (*p* = 0.02). These findings show that *T. gondii* should be considered in control measures for any outbreak or sporadic cases of abortions in goat herds.

The role of sampling time and season was also found to be significantly related to the occurrence of *T. gondii* in goats (*p* = 0.004 and *p* = 0.005, respectively). Region was also significantly linked to a higher prevalence of goat anti-*T. gondii* IgG antibodies, where goats in coastal regions were 1.7 times more exposed to the parasite than goats in plateau and mountain regions (*p* = 0.03). *T. gondii* seroprevalence in goats was higher in autumn (74.55%; 95% CI: 53.50–101.13), followed by summer (55.67%; 95% CI: 41.82–72.64), and closely similar in winter (49.06%; 95% CI: 36.64–64.33) and spring (48.51%; 95% CI: 39.39–59.12), as shown in Table 1. The difference between these rates was highly significant (*p* = 0.005). Considering the prevalence of anti-*T. gondii* IgG in different provinces, the highest rate was recorded in El Taref (70%; 95% CI: 28.14–144.23), followed by Guelma (56.10%; 95% CI: 35.56–84.17) and Constantine (53.96%; 95% CI: 42.44–667.64), and the lowest was detected in Mila (51.85%; 95% CI: 43.62–61.19) (Figure 1), which showed more than half of the animals tested in each province had anti-*T. gondii* antibodies without significant difference. 

Goats included in the present study were intended for milk, meat, and mixed production. The seroprevalence was 54.8% (95% CI: 44.44–66.85), 54.41% (95% CI: 38.31–75.00), and 51.63% (95% CI: 42.47–62.17) in farms for milk, meat, and mixed production, respectively. Statistically, a non-significant association (*p* > 0.05) was recorded between the type of production and the seroprevalence variations. Furthermore, the results provided in Table 1 showed that herd size, size of pasture area, type of pasture area, common pastures, use of concentrate feed, watering type, location of the water trough, species on the farm, presence of other farm animals, presence of cattle, equids, poultry, rodents, and other factors had no association with disease exposure. 

Regarding multivariate analysis, no significance was found for the variables comprising physiology status of males and females, transhumance, pasture, presence of dogs, and vaccination of goats against other pathogens, showing no significant association to the occurrence of anti-*T. gondii* IgG antibodies. In addition, the generalized linear mixed model analysis showed that the main risk factors potentially associated with *T. gondii* infection in goats were pasture frequency, presence of water source in pasture, number of cats, hygiene, use of anthelminthics, proportion of abortions, number of abortions in the last year, year of sampling, region, and season (Table 2).

## 4. Discussion

The current research is the first report of *T. gondii* seroprevalence and risk factor relationships in goats from Algeria’s northeastern regions. The seroprevalence found in this investigation (53.26%) was greater than that identified in central Algeria (11.92%) and Djelfa province (13.21%), as determined by indirect ELISA and indirect fluorescent anti-body tests (IFATs), respectively [27,28]. The rate of anti-*T. gondii* antibody prevalence in goats in this study was found to be lower than that (71.74%) reported in the humid areas of Mila province using an ELISA test [20], but higher than the rate of infection (35.37%) revealed in sheep in the arid and semiarid regions of northeastern Algeria using an ELISA test [29]. According to multiple studies that have employed diverse serological tests such as ELISA, IFAT, and LAT, the overall prevalence of *T. gondii* in goats in Algeria has been shown to range from 11.92% to 71.74% with an average rate of 33.61% [21]. The current rate of *T. gondii* in this study was higher than the average prevalence in Algerian goats. These variations might be explained by the serological assays applied in these investigations, as the modified agglutination test (MAT) and ELISA showed significant agreement, and almost perfect concordance between IFAT and ELISA was observed [30]. The comparison of research is challenging in this context. Our results imply that the prevalence identified in this study may be associated with suitable climatic conditions, because the four study provinces (Mila, Constantine, Guelma, and El Taref) have similar ecological patterns characterized by a wet period during most of the year [22] that are favorable for *T. gondii* oocyst formation and survival [5]. As a result, this region in Algeria has the greatest prevalence rate of the parasite in animals [21]. In different regions of the world, prevalence rates ranging from 0% to 100% have been recorded [31]. The variations were attributed to regional customs, local traditions, residents’ lifestyles, and meteorological conditions [32]. Given the various epidemiological settings, research designs, number of samples studied, diagnostic procedures, and cut-off points used, differences between studies should be carefully assessed. 

In the current study, the goat population from northeastern Algeria had a high individual prevalence rate *T. gondii* (53.26%), and 94.44% of goat herds had at least one seropositive animal, showing a widespread distribution of this parasite, as also reported in other Mediterranean countries [33,34,35]. This individual rate was higher than that documented by many researchers from different areas worldwide: 38.28% in Egypt [36], 8.5% in Morocco [37], and 34.4% in Tunisia [38]. Al-Mabruk [39] reported an exposure percentage of 71% in Libyan sheep, which is higher than the current study’s results. A total of 53.15% of goats’ blood sera were found to be positive in Faisalabad, Pakistan [40]; that was in close correlation with what we found. Gazzonis et al. [41] showed that 41.7% of tested goats in northern Italy were *T. gondii* seropositive with a comparable herd level (96.6%). Additionally, in comparison to the present data, goats showed lower herd seroprevalences of 60% and 72.2% in Morocco [37] and southern Spain [42], respectively. The seropositivity variance between these studies might be related to differences in management strategies, biosecurity measures, and climatic conditions at each goat farm [7].

The findings of this study contribute to the body of information about the epidemiology of *T. gondii* on a global scale. In many regions of the world, a few studies have looked into seroprevalence and associated risk factors in traditional husbandry systems [43]. In the current study, gender was not a notable significant risk factor for exposure to *T. gondii*, consistent with findings from Martínez-Rodriguez et al. [43] and Jilo et al. [44], which showed goats of both genders were equally likely to contract a *T. gondii* infection. The increased incidence of *T. gondii* antibodies in female goats in this study might be attributable to the fact that there were more females studied than males. Our findings revealed that animals of all ages had been exposed to the parasite in the same way, according to what was reported in previous studies [45,46] showing that age had no relationship with *T. gondii* exposure in goats. The variable “geographic areas” has a considerable influence on *T. gondii* prevalence, particularly in regions with ecological patterns typified by a long wet period of the year, which is compatible with what was reported previously [28,43,47]. This state might be due to topography and climatic differences related to geographical characteristics, rainfall, and yearly average temperatures that can promote the survival and sporulation of *T. gondii* oocysts [28]; thus, their transmission to goats in pastures is enhanced [48,49].

Different breeding strategies had no effect on *T. gondii* seroprevalence levels registered in the present investigation. These findings corroborated those published by Mohamed-Cherif et al. [28] and Rizzo et al. [50]. The current investigation confirmed the findings of Rêgo et al. [51] showing that production of milk and/or meat from goat husbandry did not positively correlate with the prevalence rate of *T. gondii* infection. Contrary to what we found, the positive rate of this parasitic infection was significantly impacted by more dairy and meat purposes in goat farming [52]. Most studies found that semi-intensive and extensive practices make it easier for goats to be exposed to *T. gondii* oocysts in the environment [38,51]. Dahmane et al. [20] reported that the degree of exposure to the parasite in Mila province, northeastern Algeria, was unaffected by herd practices. Traditional husbandry is still the most common way of raising goats in northeastern Algeria, and it might be a significant risk factor for *T. gondii* infection. A few goats are often kept by each rural family as an alternative form of animal husbandry. Early in the morning, these animals are moved from the settlements to their natural pastures, and they return in the late afternoon. The surroundings are home to a large number of stray and domestic cats; thus, there is a higher chance that these animals will contract oocysts from cat feces. 

Several reports have shown a negative correlation between flock size and *T. gondii* seroprevalence [34,41], and that was not recorded in the present study. The same outcome of *T. gondii* infection in herds with small and large numbers of animals was reported in other studies [29,42,53]. In addition, common pastures did not appear as a risk factor; thus, contact with other herds of goats or other livestock animals had no effect on *T. gondii* exposure. A non-significant correlation was observed between the *T. gondii* seroprevalence and the presence of other animal species, including sheep, cattle, poultry, and cats, which had been in close contact with goat flocks. In contrast, Udonsom et al. [45] reported that the presence of other domestic animals on a goat farm increased by 1.69 times the risk of exposure to this parasite. According to the current results, the presence of dogs appeared to be related to an increased risk of *T. gondii* exposure, and dogs may be a possible contamination source for grazing pastures [43,54]. However, other studies did not find an association between *T. gondii* infection and the presence of dogs and wild dogs [34,51]. Similarly to our findings, the presence of equines and cattle on the same farms as goats was regarded as a protective factor, decreasing goat exposure to *T. gondii* [43]. Contact with rodents was considered a potential risk factor in various studies [17]. However, this was not consistent with our findings. Overall, the presence of other animal species may have a direct effect on the risk of infection for livestock animals, including goats, where they may serve as *T. gondii* reservoirs and play a role in the expansion within the farm [41,51].

Poor management and hygienic standards characterizing goat herds in the studied locations promote the exposure of goats to *T. gondii*, particularly where herds had poor hygienic status, which was also reported previously [50,53]. Measures of hygiene and regimes of cleaning and disinfection applied at the farms may play an important role in reducing oocyst contamination [17]. Biosecurity measures such as quarantine and having parturition space for goat females had no statistically significant effect on *T. gondii* seropositivity in the present study, and these variables could be considered protective factors, which was consistent with findings from other studies [34,50]. The absence of facilities and supervised areas for delivery of females made goats more exposed to the parasite [43]. In addition, in farms where elimination of placental tissues and abortion products was not applied correctly, the risk of exposure to *T. gondii* was notably high, which might be related to the easy eating of the aborted fetuses by domestic and wild cats [51]. In our study, no association was found between *T. gondii* seropositivity and abortion tissue elimination, which might be related to the limited access of the cats to abortion products that were mostly disposed of by owners or consumed by dogs.

Consistent with previous studies [51,55], anthelmintic treatment and deworming were shown to reduce the rate of *T. gondii* infection in goat herds and protect against toxoplasmosis. In addition, the use of antiparasitic drugs against ectoparasites such as ticks and the use of antibiotics did not have a correlation with the high prevalence of *T. gondii*. However, vaccination of goats against some microbial diseases, such as enterotoxemia and brucellosis, could affect the level of exposure to the parasite because animals could become more susceptible to the infection when the immune system is in a critical state and producing antibodies against vaccines and other pathogens, such as *T. gondii*, at the same time. Stress on the immune system caused by various states of illness may be a contributing factor to the greater seroprevalence of *T. gondii* [56].

The prevalence of *T. gondii* in our study was insignificantly higher in farms with a history of reproductive disorders, as recorded by other authors [33,34,42]. It seems there was a direct relationship between the history of abortion and the seropositivity rate of *T. gondii* [29,57]. Moreover, the proportion of abortions and number of aborted fetuses in the last year were in significant correlation with the occurrence of *T. gondii*, consistently with the findings of Benlakehal et al. [29] and Gharekhani et al. [58], whereas an insignificant association was reported previously [34,42,54]. Several studies have used serological methods to identify reproductive losses associated with *T. gondii* [8]. It would be important to use a combination of diagnostic methods, such as histopathological and molecular assays, to identify any association between *T. gondii* and placental lesions, and to rule out other infectious causes of reproductive disorders [59,60]. In a recent study, the occurrence of *Coxiella burnetii* in the local goat population in the same study areas was reported [61], which reflects an exposure of this population to various abortifacient agents, and it should be taken into consideration when applying strategies to prevent and reduce the rate of abortions in goat herds.

The presence of cats on goat farms/flocks appeared to have a considerable influence on *T. gondii* prevalence and transmission [44,45], contrary to the current findings and to what was recorded in other investigations [34,41,42]. According to our results, the risk of contracting the infection increased not only with the presence of cats but mostly with their number on goat farms or in neighboring areas, which played an important role in maintaining parasite dissemination [43,62]. Interestingly, the presence of cats on grazing sites, which allows a contamination of pastures or water sources, might be responsible for the high prevalence rate, which can also be related to transhumance and rearing frequency. It was estimated that at any given time and in a given cat population, 1% to 2% of cats excrete millions of oocysts over a short period of 1 to 2 weeks throughout their lives, which ensures major contamination of the environment [1,3]. In addition, the presence of wild felids had no notable influence on *T. gondii* seropositivity, and this might be related to the limited presence of wild felids or the fact that their presence was not correctly noticed by farmers. The main cat-related risk factor identified was the number of cats. Unexpectedly, Rêgo et al. [51] found that the proximity of cats to a water source had a statistically beneficial impact. The possibility of cats contaminating cropland, feed, or water delivered to livestock appeared to be a potential risk factor that requires more investigation [17]. Cat-related factors, such as the number of cats on the farm, the contact of animals with cats or cat feces, and the contact of cats with water or water sources, promote the exposure of goats to *T. gondii* oocysts. Multivariate analysis revealed that the probability of contracting a *T. gondii* infection increased with the number of cats or their density, which was similar to a previous report [63]. In addition, the absence of feline population control programs, especially in the studied rural areas, could result in more cases of infection on goat farms [64]. Similarly to our findings, Rizzo et al. [50] showed a non-significant association between cats’ access to food storage and the rate of anti-*T. gondii* antibodies. On the other hand, access of cats to exterior water sources significantly increased oocyst exposure, and water from rivers and the public supply enhanced the probability of exposing animals to the parasite [34,41]. Furthermore, the use of water from a source (weir or dam) in containers (inside and outside the farms) did not affect the rate of *T. gondii* seropositivity [45,50], which was also reported in our study. In the study areas, the high rate of goat toxoplasmosis might be attributed to the presence of water sources in pastures such as stagnant water or lakes, which are able to be accessed by cats and eventually become contaminated with *T. gondii* oocysts. The use of stagnant water was documented as a major risk factor [62]. Other authors reported that the use of a lake’s water or surface waters appears to be more susceptible to easily transmitting the parasite compared with other water sources [28,45,65]. However, drinking water from cisterns and taps seems less able to enhance the risk of *T. gondii* transmission compared with the use of aqueduct water [43,45]. We suggested that cats can contaminate unprotected water sources such as artesian wells and small lakes in pastures, especially in rainy periods, and that makes the parasite able to complete their biological cycle in the intermediate hosts, including goats [1]. Additionally, water sources found in pastures could be contaminated by feces containing *T. gondii* oocysts excreted by pet cats in rearing areas near farms or by feral cats in forest areas far from the farm location. Overall, it is hard to quantify the risk of infection of the animals through contaminated water [17].

## 5. Conclusions

For the first time, our study showed a considerable seroprevalence of *T. gondii* infection in goats across different agro-geographical areas in northeastern Algeria, reflecting a large distribution of the parasite, which was related to factors significantly impacting the infection rate, such as the number of cats, seasonal pastures, presence of water sources in these pastures, and hygiene conditions. Control measures to prevent and reduce the transmission of *T. gondii*, including preventing access of cats and other biological or mechanical vectors for this pathogen to water sources and feed storage facilities, need to be planned. A significant association between abortions and *T. gondii* seropositivity was reported, which makes it important to apply strategies for reducing abortion rates in goat herds. 

The high prevalence of *T. gondii* in goats showed the zoonotic importance of this parasite through consumption of their meat, raw milk, and milk products; thus, it is necessary to sensitize goat owners and other peoples, through education, on the zoonotic transmission of *T. gondii*. Further epidemiological investigations based on serological and molecular assays and genotypic characterization of the parasite in goat meat designed for human consumption should be conducted to better understand the impact of caprine toxoplasmosis on food production and public health in Algeria.

## Figures and Tables

**Figure 1 animals-14-00883-f001:**
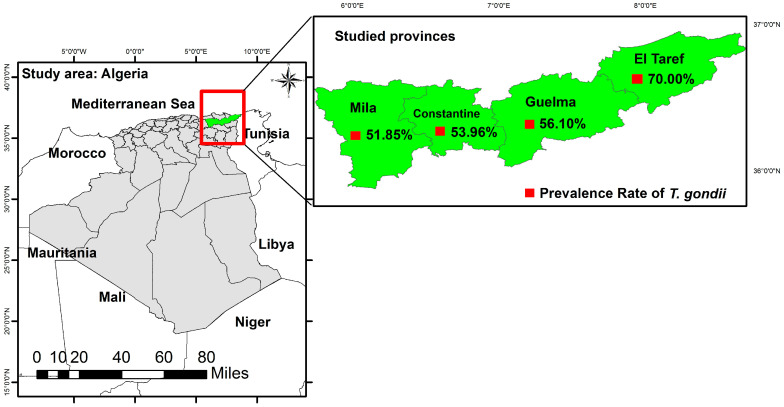
Map of the study area and prevalence rate of *Toxoplasma gondii* in goats submitted to the epidemiological investigation by province.

**Table 1 animals-14-00883-t001:** Seroprevalence and univariate analysis of explanatory risk factors associated with *T. gondii* infection in the local goat population from northeastern Algeria.

Variable	Categories	Negative	Positive (%)	OR (95% CI)	*p*-Value
**Individual-related factors**					
Physiology status of male	Breeder	24	43 (64.18)	2.0903 (1.0939–3.9941)	0.024
	Non-breeder	49	42 (46.15)		
Physiology status of female	Pregnant	46	59 (56.19)	0.2599 (0.0786–0.859)	0.020
	Non-pregnant	12	4 (25)		
**Herd-related factors**					
Pasture	No	7	20 (74.07)	0.3786 (0.1569–0.9138)	0.025
	Yes	208	225 (51.96)		
Pasture frequency	Never	7	20 (74.07)	Ref	0.001
	Sporadic	57	93 (62.00)	1.7512 (0.6967–4.4018)	
	Seasonal	124	113 (47.68)	1.7904 (1.1802–2.7161)	
	Frequent	27	19 (41.30)	1.295 (0.6829–2.4556)	
Transhumance	No	206	242 (54.02)	0.2837 (0.0758–1.062)	0.046
	Yes	9	3 (25)		
Presence of water source in pasture	No	163	139 (46.03)	2.2411 (1.4641–3.4304)	0.0001
	Yes	45	86 (65.65)		
Presence of dogs	No	27	50 (64.94)	0.5601 (0.3366–0.932)	0.024
	Yes	188	195 (50.91)		
**Cat-related factors**					
Number of cats	1–2	10	03 (23.08)	Ref	0.031
	3–4	94	125 (57.08)	4.4326 (1.1869–16.5548)	
	˃5	64	66 (50.77)	0.7755 (0.5016–1.199)	
**Hygiene-related factors**					
Hygiene	Bad	53	49 (48.08)	Ref	0.004
	Good	39	23 (37.10)	1.5677 (0.8223–2.9889)	
	Medium	123	173 (58.45)	2.3849 (1.3558–4.1952)	
**Disease- and herd-health-related factors**					
Use of anthelminthics	No	133	178 (57.23)	0.6105 (0.412–0.9047)	0.013
	Yes	82	67 (44.97)		
Vaccination of goats against other pathogens	No	98	88 (47.31)	1.4944 (1.0277–2.1729)	0.035
	Yes	117	157 (57.3)		
**Reproduction-related factors**					
Proportion of abortions	0%	33	19 (36.54)	Ref	0.003
	1–20%	130	136 (51.13)	0.5504 (0.298–1.0165)	
	21–50%	49	83 (62.88)	0.6176 (0.4029–0.9467)	
Number of abortions in the last year	0	33	19 (36.54)	Ref	0.022
	1–5	162	205 (55.86)	0.455 (0.2495–0.8298)	
	6–10	17	14 (45.16)	1.5366 (0.7355–3.2102)	
**Spatio-temporal-related factors**					
Year of sampling	2020	29	57 (66.28)	Ref	0.004
	2021	81	76 (48.41)	2.0948 (1.2138–3.6155)	
	2022	94	109 (53.69)	0.8092 (0.5331–1.2281)	
	2023	11	3 (21.43)	4.2518 (1.1517–15.6964)	
Region	Coastal	3	7 (70.00)	Ref	0.031
	Plateau	109	149 (57.75)	1.7069 (0.4316–6.7504)	
	Mountain	103	89 (46.35)	1.582 (1.086–2.3044)	
Season	Autumn	14	41 (74.55)	Ref	0.005
	Winter	54	52 (49.06)	3.0412 (1.4858–6.225)	
	Spring	104	98 (48.51)	1.0219 (0.6385–1.6355)	
	Summer	43	54 (55.67)	0.7504 (0.4613–1.2205)	

Ref.—reference value; OR—odds ratio; CI—confidence interval.

**Table 2 animals-14-00883-t002:** Seroprevalence of *T. gondii* antibodies and multivariate linear regression analysis of potential risk factors for local goat populations from northeastern Algeria.

Variable	Category	SE	OR	95% CI	*p*-Value
Pasture frequency	Never		Ref		
	Sporadic	0.983	1.806	1.241–2.630	0.547
	Seasonal	1.105	3.804	3.321–4.358	0.003
	Frequent	1.203	2.756	2.320–3.149	0.036
Presence of water source in pasture	Yes		Ref.		
	No	0.466	4.844	1.942–7.789	0.0004
Number of cats	1–2		Ref.		
	3–4	1.613	2.364	1.000–4.237	0.004
	˃5	1.968	3.059	1.450–6.879	0.031
Hygiene	Bad	0.620	4.902	1.653–5.574	0.003
	Good	-	Ref		-
	Medium	0.665	4.289	1.581–5.829	0.035
Use of anthelminthics	Yes	0.638	2.640	1.592–3.146	0.036
	No		Ref		
Proportion of abortions	0%	-	Ref		
	1–20%	0.703	0.891	0.647–1.265	0.025
	21–50%	1.027	1.450	1.023–2.845	0.004
Number of abortions in the last year	0		Ref		
	1–5	2.392	4.641	4.181–6.142	0.005
	6–10	2.654	4.964	3.552–5.797	0.008
Year of sampling	2020		Ref		
	2021	0.740	1.814	0.740–3.264	0.046
	2022	0.892	1.036	0.843–2.631	0.038
	2023	1.082	1.914	1.237–3.816	0.022
Region	Costal		Ref		
	Plateau	0.721	2.157	1.450–3.641	0.0015
	Mountain	0.569	3.658	1.197–4.919	0.0001
Season	Autumn		Ref		
	Winter	1.158	2.120	1.814–3.761	0.023
	Spring	1.648	1.846	1.023–3.460	0.005
	Summer	1.271	0.951	0.698–2.024	0.020

SE: standard error; Ref—reference value; OR—odds ratio; CI—confidence interval.

## Data Availability

The data presented in this study are available on request from the corresponding author. The data are not publicly available due to [confidentiality agreements with participants/subjects involved in the research].

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
