# Peer review of "Seroprevalence Assessment and Risk Factor Analysis of Toxoplasma gondii Infection in Goats from Northeastern Algeria"

_animals, 2024, doi:10.3390/ani14060883_

Round 1

Reviewer 1 Report

Comments and Suggestions for Authors

The authors sought to conduct a cross-sectional epidemiological study based on random samples for toxoplasmosis in goats in four provinces of Algeria to determine risk factors. The study design presented in the manuscript is scientifically sound and was carried out rationally. The manuscript text is also fine, but the organization of the manuscript is not perfect, e.g. Introduction, methodology and results are sufficiently mentioned, but the discussion section seems too long for a standard scientific manuscript and to maintain the reader's attention. Additionally, the authors cited 116 references throughout the manuscript to support their arguments, giving the impression of a review article rather than an original survey-based research article. Meanwhile, the written abstract is relatively short. I understand that the factors analyzed are numerous and there is no limit to the number of words from the journal, but the discussion part needs to be much more concise. Conclusions should be drawn from the results.

Some specific comments:

Heading 2.2: Have you collected samples from apparently healthy goats or those with signs and symptoms?

Lines 171: What was the scientific basis for the risk factors in the structured questionnaire?

Comments on the Quality of English Language

The English language is fine.

Reviewer 2 Report

Comments and Suggestions for Authors

The manuscript is very well written and designed and deals with an important subject and theme for the region, considering that the study will be the most important report on the topic for this region of Africa, to date.

However, I understand that it will have an average citation potential, considering that it only provides the seroepidemiology of toxoplasmosis in goats as its main information. There is no additional technique, new methodology, or other innovative or new presentation. However, this does not take away the scientific merit of the study.

I understand that with a few corrections, it could be published in Animals.

Below are the suggestions, which must be followed in full:

Keywords: Do not use the same words as the title. Use synonyms or other approaches to your manuscript.

Abstract. Alternate words to increase citation potential: e.g. – use toxoplasmosis, use multivariate analysis, etc.

In abstract and results: Add 95% CI for all detected prevalences. Suggestion: prevalence of 53.26% (245/460; CI 95% XX,X-XX,X).

In Figure 1 I indicate this means the percentages.

Line 125: Informs the annual average temperature, annual relative humidity and annual rainfall according to the literature cited

Item 2.2: inform the expected error.

L: 172-173: State the literature used to choose these variables. Cite these articles.

L 178: 2000xg

2.4: State what the positive and negative controls are. State where it was used, whether from a batch of serums or a diagnostic sample. Cite the origin.

Table 1: Leave only p values <0.05. Make a table, and add 1 as supplementary material, with all univariate analyses with a p-value ≥ 0.05. You can indicate in the text that these results are available in Supplementary Material 1 and/or also inform these results in the text.

Round 2

Reviewer 1 Report

Comments and Suggestions for Authors

The manuscript has been revised but I still have some comments: As far as I observed, the authors didn't address my previous questions The authors did reduce the number of references significantly but still I could find many references that were not directly related to the author's argument e.g. citing many references from the countries that aren't directly related to Algeria- doesn't make any sense. Moreover, the discussion text is still too long for a standard research article, especially in the scenario how other sections of the manuscript are organized. Furthermore, the conclusions should be clearly written as a take-home message e.g. just for the author's idea, I wrote this section for the authors but please don't copy and paste it directly in the revised manuscript. Moreover please mention how the genotyping is possible for the T. gondii is possible in a resource-limited country like Algeria? Moreover, the authors didn't specifically address the importance of biosecurity and watering sources in the manuscript in the discussion:

Our study indicated a considerable seroprevalence of T. gondii in goats across northeastern Algeria. Factors such as the presence of cats, seasonal pastures and grazing patterns, water sources in these pastures, and hygiene conditions significantly impacted seroprevalence incidence. Control measures to prevent T. gondii transmission included strict biosecurity i.e. preventing cats and other biological or mechanical vectors for such pathogens at farms, watering sources, and feed storage areas. A significant correlation between T. gondii seropositivity and goat abortions was found, indicating a need for strategies to mitigate abortion rates in these goat herds which can only be achieved by addressing and eliminating the associated factors. This significant correlation also pointed out the zoonotic importance of this parasite through the consumption of meat, raw milk, and milk products of these goats, highlighting the need of public awareness and education on T. gondii zoonosis. That’s why our study called for further in-depth epidemiological investigations, including serological and molecular assays, and genotypic characterization of T. gondii (isolation of the pathogen-please specify how?), to better understand caprine toxoplasmosis's impact on livestock and public health in Algeria.

Comments on the Quality of English Language

Minor corrections are needed.
